# Glucose Uptake Is Increased by Estradiol Dipropionate in L6 Skeletal Muscle Cells

**DOI:** 10.3390/ph16010025

**Published:** 2022-12-25

**Authors:** Yanhong Yao, Xinzhou Yang, Jinhua Shen, Ping Zhao

**Affiliations:** 1Hubei Provincial Key Laboratory for Protection and Application of Special Plants in Wuling Area, Institute for Medical Biology, College of Life Sciences, South-Central Minzu University, Wuhan 430074, China; 2Hubei Medical Biology International Science and Technology Cooperation Base, Wuhan 430074, China; 3School of Pharmaceutical Sciences, South-Central Minzu University, Wuhan 430074, China

**Keywords:** Estradiol Dipropionate, glucose transporter 4, Ca^2+^, L6 skeletal muscle cells, type 2 diabetes

## Abstract

GLUT4 is an important glucose transporter, which is closely related to insulin resistance and type 2 diabetes. In this study, we investigated the mechanism of Estradiol Dipropionate (EDP) on uptake of glucose in L6 skeletal muscle cells. In our study, we confirmed that EDP promoted uptake of glucose in L6 skeletal muscle cells in both normal and insulin resistant models. Western blot indicated that EDP accelerated GLUT4 expression and significantly activated AMPK and PKC phosphorylation; the expression of GLUT4 was significantly inhibited by AMPK inhibitor compound C and PKC inhibitor Gö6983, but not by Wortmannin (Akt inhibitor). Meanwhile, EDP boosted GLUT4 expression, and also increased intracellular Ca^2+^ levels. In the presence of 2 mM, 0 mM extracellular Ca^2+^ and 0 mM extracellular Ca^2+^ + BAPTA-AM, the involvement of intracellular Ca^2+^ levels contribute to EDP-induced GLUT4 expression and fusion with plasma membrane. Therefore, this study investigated whether EDP promoted GLUT4 expression through AMPK and PKC signaling pathways, thereby enhancing GLUT4 uptake of glucose and fusion into plasma membrane in L6 skeletal muscle cells. In addition, both EDP induced GLUT4 translocation and uptake of glucose were Ca^2+^ dependent. These findings suggested that EDP may be potential drug for the treatment of type 2 diabetes.

## 1. Introduction

With the rapid development of economic level in the world, diabetes mellitus (DM) is a disease that seriously endangers human health, of which type 2 diabetes mellitus (T2DM) accounts for more than 90%. Skeletal muscle is one of the largest organs in the human body and plays an important role in movement and systemic metabolic homeostasis. The formation of muscle fibers during myogenesis and muscle repair involves the activation of progenitor cells, which multiply in the form of mononuclear myoblasts and eventually fuse to form multinucleated myotubules. This complex arrangement limits the methods that can be used to study the processes within myoblasts in vivo [1]. However, molecular exploration can be gained from in vitro models of skeletal muscle, such as L6 cells isolated from a newborn rat thigh muscle culture [2]. Glucose transporter 4 (GLUT4) is one of the 13 facilitative glucose transport proteins encoded in the genome, and is most abundant in adipose tissue, heart, and skeletal muscle [3]. Glucose uptake and glycogen synthesis are particularly important in skeletal muscle, which is responsible for processing more than 70% of systemic glucose [4]. Therefore, reduced responsiveness of skeletal muscle to insulin, known as insulin resistance, is an important aspect in the development of type 2 diabetes mellitus (T2DM) [5].

In normal conditions, cellular glucose uptake is facilitated by insulin release with the help of glucose transporter (GLUT4) to the cell’s surface [6]. We observed significantly elevated activation and phosphorylation of IRS-1, phosphorylation of serine Akt, phosphorylation of threonine Akt, and phosphorylation of threonine AS160 in diet-induced T2DM adipocytes [7]. Altered GLUT4 translocation results in disrupted uptake of glucose, leading to insulin resistance [8]. In addition, downstream of Adenosine 5’-monophosphate (AMP)-activated protein kinase (AMPK), contraction induces phosphorylation of the Rab-GAPs AS160 and TBC1 domain family member 1 (TBC1D1) to regulate GLUT4 vesicular trafficking stimulated by insulin or contraction [9,10] and protein kinase C (PKC) regulates contractile induced GLUT4 transport in skeletal muscle cells [11]. TBC1D4 (also known as Akt substrate of 160 kDa, hence AS160) is a major manager of insulin-dependent GLUT4 trafficking to the plasma membrane, as confirmed by studies conducted on skeletal muscle and adipocyte tissue [12,13]. AS160 and TBC1D1 are known as Rab-GTPase activating proteins (GAP) mediating GLUT4 traffic. In basal condition, AS160 and TBC1D1 have Rab GAP activity to maintain their Rab substrates in the inactive GDP-bound form which hold GLUT4 retention. Upon insulin challenge, AS160 and TBC1D1 become phosphorylated and their Rab GAP becomes inactive. Their inhibition on Rabs is removed. The activated Rabs regulate GSV moving to cell surface [14].

Nowadays, monomeric compounds have played an important role in the treatment of diabetes such as glimepirideas well as insulin. EDP is a semisynthetic steroidal estrogen used for hormonal therapy of menopausal symptoms and low estrogen levels in women and also for the treatment of gynecological diseases. There are many published data demonstrating the value of estrogen therapy in moderating menopausal symptoms such as insomnia and hot flashes and in protecting osteoporosis and urogenital atrophy [15]. In the previous work, we found a hypoglycemic monomer drug screened from the L6 cell diabetes model. In the initial preliminary experiment, we found that EDP could increase the glucose uptake in L6 skeletal muscle cells through the glucose detection kit, and MTT results showed that EDP had no cytotoxic effect in L6 skeletal muscle cells. Subsequently, EDP could increase GLUT4 expression and fuse into plasma membrane by laser confocal microscopy. Therefore, it is speculated that EDP may have hypoglycemic activity, so as to explore its functional experiments. The hypoglycemic activity of EDP has not been studied or reported so far. We demonstrated that EDP promotes GLUT4 expression through activating phosphorylation of PKC and AMPK, thereby enhancing GLUT4 fusion into the plasma membrane and uptake of glucose. Meanwhile, we verified that both EDP-promoted uptake of glucose and induced GLUT4 fusion into the plasma membrane were Ca^2+^ dependent in L6 skeletal muscle cells. These studies show that EDP can be used as a potential drug for the therapy of type 2 diabetes.

## 2. Results

### 2.1. EDP Promotes Uptake of Glucose in Normal and Insulin-Resistant L6 Cells

In the present study, EDP with hypoglycemic activity was previously selected among a large number of small molecule compounds. Figure 1A shows the chemical structure of EDP. The ability of EDP to stimulate uptake of glucose in L6 skeletal muscle cells was explored by setting insulin as a positive drug. From Figure 1B, EDP promoted uptake of glucose in L6 cells after treatment of cells with different concentrations of EDP, respectively. Under the condition of insulin resistant model [16], EDP also promoted uptake of glucose, while the uptake of glucose ability of insulin was significantly weakened (Figure 1C). The EDP was found to be non-toxic to cells as measured by MTT assay (Figure 1D).

### 2.2. EDP Increases Intracellular GLUT4 Expression

Next, we explored the effect of EDP on GLUT4 expression in L6 cells by confocal laser scanning microscopy. L6 cells were transfected with the lentivirus vector GV348-myc-GLUT4-mOrange, which encodes a mOrange fusion protein of GLUT4 tagged with myc epitopes [17]. Gradually increasing over time, GLUT4 expression induced by 10 μΜ EDP after 30 min treatment increased by approximately 2.3-fold (Figure 2A,B). In addition, the expression level of GLUT4 protein was verified using Western blot. Next, we examined the changes of GLUT4 at different times and different concentrations. The time gradient diagram of EDP promoting GLUT4 expression is shown in Figure 2C, GLUT4 expression was about 1.38-fold and 1.55-fold at 5 and 30 min after insulin treatment, and EDP increased the expression of GLUT4 by 1.52-fold and 1.62-fold. Similarly, we found that GLUT4 protein expression level increased in L6 skeletal muscle cells after 30 min of treatment with different concentrations of EDP (Figure 2D). The results revealed that EDP increases GLUT4 expression in L6 cells.

### 2.3. EDP Enhanced GLUT4 Expression and Fusion into the Plasma Membrane in L6 Cells

GLUT4 is the major insulin-dependent transporter on skeletal muscle cells membranes and its protein expression level determines its ability to transport glucose. The expression of GLUT4 induced by EDP was detected in skeletal muscle cells by laser confocal scanning microscopy and Western blot (Figure 2). We then wondered if EDP would promote GLUT4 fusion into the plasma membrane. When insulin and EDP were added to stimulate for 30 min, the expression of intracellular GLUT4 was significantly enhanced (red fluorescence) compared with the negative group, indicating that the effect is to produce more GLUT4, allowing increased GLUT4 vesicles to be transported to the cell surface and fuse into the plasma membrane (FITC fluorescence on the cell surface is significantly enhanced). At the same time, the effect of EDP is equivalent to that of insulin (Figure 3A). GLUT4-mOrange positive cells were enhanced about 5.2-fold with insulin and 4.8-fold with EDP (Figure 3B). Meanwhile, the proportion of FITC fluorescent cells in GLUT4-mOrange positive cells was 58% for insulin and 55% for EDP (Figure 3C).

### 2.4. EDP Promotes Expression of GLUT4 Mainly through AMPK and PKC Signaling Pathways

PKC, AMPK, and PKB/Akt protein were important in regulating GLUT4 expression and translocation. Western blot results showed that the concentration gradients of EDP promote AMPK and PKC phosphorylation (Figure 4A,C). After stimulation with phorbol 12-myristate 13-acetate (PMA) [18], insulin [19], and metformin [20] as the positive control, the phosphorylation levels of PKC, Akt, and AMPK were distinctly enhanced. However, EDP-induced GLUT4 expression did not activate phosphorylation of Akt signaling pathways (Figure 4B). Then reverse validation was performed to verify whether EDP activated the expression of GLUT4 through these three pathways. It was found that GLUT4 expression was also decreased in the presence of PKC inhibitor Gö6983 and AMPK inhibitor compound C. However, wortmanin, an Akt inhibitor, did not affect the expression of GLUT4 (Figure 4D). These results showed that EDP promoted GLUT4 expression mainly through activation of AMPK and PKC pathways.

### 2.5. EDP Enhanced Intracellular Ca^2+^ Concentration in L6 Skeletal Muscle Cells

Next, we further explored intracellular second messenger Ca^2+^, which is a major player in cellular metabolism, and therefore explored the molecular mechanism of GLUT4 expression as well as transport regulation by EDP. The myc-GLUT4-mOrange-L6 cells were stably transfected with lentivirus, where GLUT4-mOrange expression was marked in red and Ca^2+^ was marked green with Fluo-4 AM dye. After stimulation with EDP for 30 min, GLUT4-mOrange red fluorescence was significantly enhanced by approximately 2.6-fold within 30 min (Figure 5A,B), accompanied by green fluorescence with intracellular accumulation of large amounts of Ca^2+^. Confocal laser scanning microscopy quantization showed that Fluo-4 AM-labeled Ca^2+^ fluorescence also enhanced by approximately 10.3-fold within 30 min (Figure 5C). These results conjectured a hypothesis that Ca^2+^ is involved in facilitating EDP-mediated GLUT4 expression.

### 2.6. Engagement of Ca^2+^ Favors EDP-induced of Uptake of Glucose and GLUT4 Fusion into the Plasma Membrane

Subsequently, the effect of EDP-induced uptake of glucose was further verified in the presence of 2 mM, 0 mM extracellular Ca^2+^, and 0 mM extracellular Ca^2+^ + 10 μM BAPTA-AM. BAPTA-AM is an intracellular Ca^2+^ chelator that blocks the enhancement of Ca^2+^ concentration [9,21]. We observed that EDP promoted uptake of glucose under the condition of 2 mM extracellular Ca^2+^, while EDP-stimulated uptake of glucose was weakened in the presence of 0 mM extracellular Ca^2+^, the reduction of EDP-stimulated uptake of glucose was more obvious in the presence of 0 mM extracellular Ca^2+^ + 10 μM BAPTA-AM, indicating that Ca^2+^ had a great influence on uptake of glucose (Figure 6).

Next, the dynamics of fluorescence in myc-GLUT4-mOrange-L6 cells were detected by immunofluorescence assay in the presence of 2 mM, 0 mM extracellular Ca^2+^, and 0 mM extracellular Ca^2+^ + 10 μM BAPTA-AM. Similarly, compared with the control group, insulin and EDP-induced GLUT4 fusion with plasma membrane was significantly enhanced (FITC fluorescence was significantly enhanced) and GLUT4 expression was also enhanced (GLUT4-mOrange fluorescence was enhanced) in 2 mM extracellular Ca^2+^. However, EDP-induced GLUT4 expression was inhibited by 0 mM extracellular Ca^2+^ and the inhibition was particularly pronounced by 0 mM extracellular Ca^2+^ + 10 μM BAPTA-AM, resulting in reduced GLUT4 fusion with plasma membrane. When both intracellular and extracellular Ca^2+^ were restricted, GLUT4 expression and fusion with the plasma membrane were correspondingly restricted, suggesting that the GLUT4 expression and fusion into plasma membrane are closely related to intracellular Ca^2+^ level (Figure 7A–C).

## 3. Discussion

In females of many species, the end of reproductive life is associated with a reduced state of estrogen, which leads to other physiological changes that increase the risk of osteoporosis and cardiovascular disease [15]. EDP is an estrogen–progestogen combination that can be used as an estrogen progestogen agonist. Estrogen deficiency is detrimental to many wound-healing processes, notably inflammation and re-granulation, while exogenous estrogen treatment widely reverses these effects. New molecular techniques, coupled with increased understanding of estrogen in skin biology, will provide further opportunities to develop estrogen receptor-targeted therapeutic [22]. There are few studies on the hypoglycemic activity of EDP. In the present study, we not only researched the hypoglycemic activity of EDP in L6 skeletal muscle cells, but also investigated the effect of EDP on GLUT4 expression, fusion into plasma membrane, and intracellular Ca^2+^ involvement.

The expression of GLUT4 is regulated by complex mechanisms [23]. GLUT4 is not only the most important glucose transporter in skeletal muscle, but GLUT4-induced glucose transport is the rate-limiting step in uptake of glucose and metabolism [24]. Muscle contraction mediates GLUT4 transport and insertion into the plasma membrane for uptake of glucose in skeletal muscle. As the primary site of insulin stimulated uptake of glucose, skeletal muscle is also considered to be a major driver of systemic insulin resistance [25,26]. It has been reported that promoting the translocation of GLUT4 to the cell membrane can enhance uptake of glucose [27,28]. In this research, we found that EDP was not only able to increase uptake of glucose in normal and insulin resistant L6 skeletal muscle cells (Figure 1B,C), but also increased EDP-induced GLUT4 expression (Figure 2A–D). In addition, EDP also increased GLUT4 fusion into the plasma membrane in L6 skeletal muscle cells (Figure 3A–C).

Next, we sought to identify signal transduction pathways related to stimulating GLUT4 expression. Previous studies have shown that the PKB/Akt, AMPKα, and PKC signaling pathways were related in the GLUT4 transport and expression [29,30,31,32,33]. The glucose transport is regulated by two different pathways in skeletal muscle. One is insulin stimulation via insulin receptor substrate 1 (IRS-1)/PI3 kinase; the other is through AMPK activation by muscle contraction [34]. EDP was found to promote GLUT4 expression through AMPK and PKC pathways, but not through the Akt pathway (Figure 4A–D).

The rapid enhance in Ca^2+^ concentration during muscle contraction could be one of the initial signals for exercise-stimulated GLUT4 biogenesis [21]. The rising extracellular Ca^2+^ level led to in 2-fold increases in GLUT4 expression in L6 myocytes and insulin stimulated glucose uptake [35]. It has been showed in the literature that the intracellular Ca^2+^ plays a significant role in insulin induced GLUT4 transduction [29,36] and uptake of glucose [37]. EDP stimulated GLUT4 expression and intracellular Ca^2+^ level (Figure 5A). Meanwhile, we found that EDP promoted uptake of glucose in the presence of the 2 mM extracellular Ca^2+^ in L6 skeletal muscle cells. However, in the presence of 0 mM extracellular Ca^2+^ and 0 mM extracellular Ca^2+^ + BAPTA-AM, the promoting effect of EDP on uptake of glucose was expressively inhibited (Figure 6). Similarly, EDP-induced GLUT4 expression and fusion with plasma membrane was expressively enhanced in the presence of the 2 mM extracellular Ca^2+^. However, EDP-induced GLUT4 expression and fusion into plasma membrane were slightly inhibited at 0 mM extracellular Ca^2+^, but almost entirely inhibited at 0 mM extracellular Ca^2+^ + BAPTA-AM (Figure 7).

In conclusion, our study indicated that EDP induces GLUT4 expression and fusion into the plasma membranes to promote uptake of glucose, and revealed that EDP promoted GLUT4 protein expression level through AMPK and PKC signaling pathways. EDP also enhanced GLUT4 expression, which was accompanied by a large accumulation of Ca^2+^ in L6 skeletal muscle cells. In addition, our studies testified that both EDP-promoted uptake of glucose and promoted GLUT4 fusion to the plasma membrane were Ca^2+^ dependent (Figure 8). Of course, this study also has some shortcomings. The hypoglycemic activity of EDP in animals needs to be further verified. In conclusion, EDP is expected to be an effective drug for the treatment of type 2 diabetes. EDP holds promise as a potential drug for type 2 diabetes.

## 4. Materials and Methods

### 4.1. Chemicals and Reagents

EDP was purchased on the official website of Selleckchem. Glucose assay kits were purchased from Intec PRODUCT INC (Xiamen, China). Mouse anti-c-myc monoclonal antibody (Cat# HT101) and anti-mouse FITC antibody (Cat# HS211-01) were ordered from TransGen Biotech (Beijing, China), the dye Fluo-4 AM (Cat# F14201) was purchased by Thermo Fisher Scientific (New York, NY, USA). Compound C (Cat# S7840) and GÖ6983 (Cat# S2911) were the products of Selleckchem (Houston, TX, USA). Wortmannin (Cat# 681676) and BAPTA-AM (Cat# 196419) were purchased from Sigma (Darmstadt, Germany). β-Actin Mouse Monoclonal Antibody (Cat# AF0003) and BCA protein concentration determination kit (Cat# P0012) were purchased from Beyotime. GLUT4 mouse antibody (Cat# 2213S), AMPKα antibody (Cat# 2532S), the rabbit phospho-AMPKα (T172) antibody (Cat# 4188S), Akt antibody (Cat# 9272S), the rabbit phospho-Akt (Ser473) antibody (Cat# 4058S), and the rabbit phospho-PKC (pan) (Thr410) antibody (Cat# 2060S) were purchased on the official website of Cell Signaling Technology (Beverly, MA, USA). Phosphate-buffered saline (PBS, Cat# CR10201S) and Tris Buffered Saline (TBS, Cat# CR10301M) were purchased on the official website of Monad (Mona, Suzhou, China). The 2 mM extracellular Ca^2+^ solution consisted of 135 mM NaCl, 5 mM KCl, 1 mM MgCl_2_, 2 mM CaCl_2,_ 10 mM HEPES, and 10 mM Glucose (pH = 7.4). There are 135 mM NaCl, 5 mM KCl, 1 mM MgCl_2_, 0.5 mM EGTA, 10 mM HEPES, and 10 mM Glucose in 0 mM extracellular Ca^2+^ solution (pH = 7.4).

### 4.2. Cultivation of L6 Skeletal Muscle Cells

Skeletal muscle myoblast cells of rat L6 were purchased from Procell Life Science & Technology Company (Cat# CL-0136, Wuhan, China). The frozen L6 cells were thawed from liquid nitrogen. The culture medium was configured by adding 10% fetal bovine serum (FBS), 1% antibiotics (penicillin 100 U/mL, streptomycin 100 μg/mL), and 89% basal medium (alpha-MEM, Gibco, Grand Island, NY, USA). The cells were cultured with 10% FBS normal medium and then replaced with 2% FBS differentiation medium for 5 to 7 d. These cells were placed at 37 °C in an incubator with 5% CO_2_.

### 4.3. Uptake of Glucose Assays

The well-grown L6 cells were seeded in 96-well plates with 10 replicates for each experimental group, and cultured with blank group, positive, and drug group, and then differentiated into myotubes for 5 to 7 d. Before the experiment, L6 cells were cultured with serum-removed medium for 2 h, and stimulated with serum-removed medium, 100 nM insulin, and different concentrations of EDP in an incubator at 37 °C and 5% CO_2_ for 0.5 h. After the drug EDP acted on L6 cells for 30 min, it reacted with glucose oxidase to produce red quinone mimide pigment, and red quinone imide pigment was proportional to the concentration of glucose in the sample, so absorbance measurement could be performed at a wavelength of 505 nm. Then, another 96-well plate was replaced to absorb the corresponding supernatant and glucose standard. In addition, glucose oxidase reagent was added to each well with 200 μL, and the absorbance value was measured by microplate reader with a wavelength of 505 nm. To detect cytotoxicity, the medium was removed after the completion of the drug effect, and 0.5 mg/mL MTT solvent was added to each well and incubated for 4 h in the dark. The waste liquid was poured out, and then the absorbance value at 492 nm was measured by adding DMSO solvent to each well with microplate analyzer. In the insulin resistance model, L6 cells were incubated with 1 μM high concentration of insulin for 1 d, and then the drug was added for reaction, while the control cells were unchanged. Glucose assay kits were purchased from in Tec. PRODUCT. INC. (Xiamen, China). Glucose uptake was quantified using the reagent for GLU-OX test (GOD-POD method) instructions on the official website. These data were analyzed by Graphpad prism7 software.

### 4.4. GLUT4 Expression and Fusion into Plasma Membrane Were Detected by Confocal Laser Microscopy

Myc-GLUT4-mOrange-L6 cells [33] were seeded onto coverslips and 10% FBS medium was added. One day later, the cells were differentiated and cultured in 2% FBS for 5 to 7 d. First, L6 cells were cultured with serum-removed medium for 2 h, followed by incubation with insulin and EDP for 0.5 h. In addition, the cells were placed in a shaker for 30 min with 3% paraformaldehyde (Servicebio, Wuhan, China) and blocked in 5% bovine serum albumin (BioFroxx, Guangzhou, China) in PBS solution for 1 h, and then incubated with anti-myc antibody for 1 h and goat anti-mouse FITC for 1 h. Finally, dynamic change of fluorescence of the cells on the glass covering was monitored using laser scanning confocal microscopy (LSM 700; Carl Zeiss, Jena, Germany). The analysis of image and the number of cells carrying FITC green fluorescence were calculated by ZEN 2010 software (Zeiss, Gottingen, Germany).

### 4.5. Western Blot Analysis

The differentiated L6 cells were in serum-removed medium for 2 h. L6 cells were slowly washed with pre-cooled PBS and solubilized with RIPA lysate (Cat# P0013B, Beyotime, Shanghai, China) containing PMSF protease inhibitor cocktail (Cat# ST506, Beyotime, Shanghai, China) and phosphatase inhibitor cocktail (Cat# B15001, bimake, Shanghai, China) for 30 min on ice. The mixture was centrifuged at 12,000 rpm at 4 °C for 15 min, and the supernatant absorbed was total cellular protein. The lysate was added relative to SDS-PAGE (Cat# P0015L, Beyotime, Shanghai, China) protein loaded buffer, and denatured on a metal heater at 95 °C for 10 min. These separated proteins were electrophoretically transferred to a nitrocellulose filter (Pall, Shanghai, China) membrane. Then the nitrocellulose filter membrane was blocked with 5% skim milk (Cat# GC310001, Servicebio, Wuhan, China) in TBS with Tween-20 (BioFroxx, Guangzhou, China) solution for 2 h. Lastly, the NC membranes were incubated with the first antibodies overnight and the second antibodies for 1 h. The ChemiDoc XRS (Bio-Rad, CA, USA) was used to image, and the gray value of protein was calculated by Image J software.

### 4.6. Real-Time Monitoring of GLUT4 Expression and Ca^2+^ Levels by Confocal Laser Scanning Microscopy

The cells were in a state of starvation before the experiment began, and incubated with 2 μM Fluo-4 AM dye in the dark about 15 min to label intracellular Ca^2+^ and then washed twice with PSS solution. The cells were monitored by LSM 700 laser scanning confocal microscope, and GLUT4-mOrange and Fluo-4 fluorescence were excited at excitation wavelengths of 555 nm and 488 nm, respectively. The laser scanning confocal microscope has an objective numerical aperture (NA) of 0.35 and an oil lens at 40X lens. The images of EDP-stimulated cells were taken 36 times for a short time, each time 10 s, and 5 min for a long time, a total of 30 min.

### 4.7. Data Analysis

The results of confocal laser scanning microscope images and the gray value of protein band were analyzed with CorelDRW, ImageJ, and GraphPad Prism 7. Differences between the two groups were analyzed by using a two-tailed Student’s *t*-test, and differences between multiple groups were analyzed by one-way ANOVA and Tukey‘s test. The data were shown as mean ± standard error, and each experiment was repeated more than three times. The difference was statistically significant when *p* < 0.05.

## Figures and Tables

**Figure 1 pharmaceuticals-16-00025-f001:**
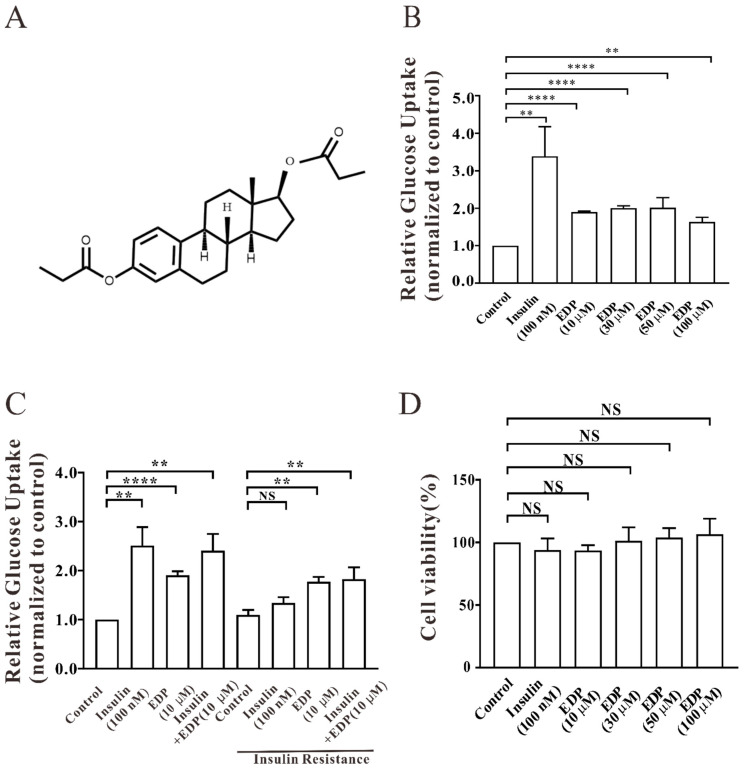
Estradiol Dipropionate (EDP) promotes uptake of glucose in L6 cells. (**A**) Chemical structure of EDP; (**B**) 10 μM EDP increases uptake of glucose in L6 cells within 30 min; (**C**) 10 μM EDP increased uptake of glucose in insulin resistant models within 30 min; (**D**) the toxicity of EDP to cells was detected within 30 min. NS *p* > 0.05; ** *p* < 0.01; **** *p* < 0.0001.

**Figure 2 pharmaceuticals-16-00025-f002:**
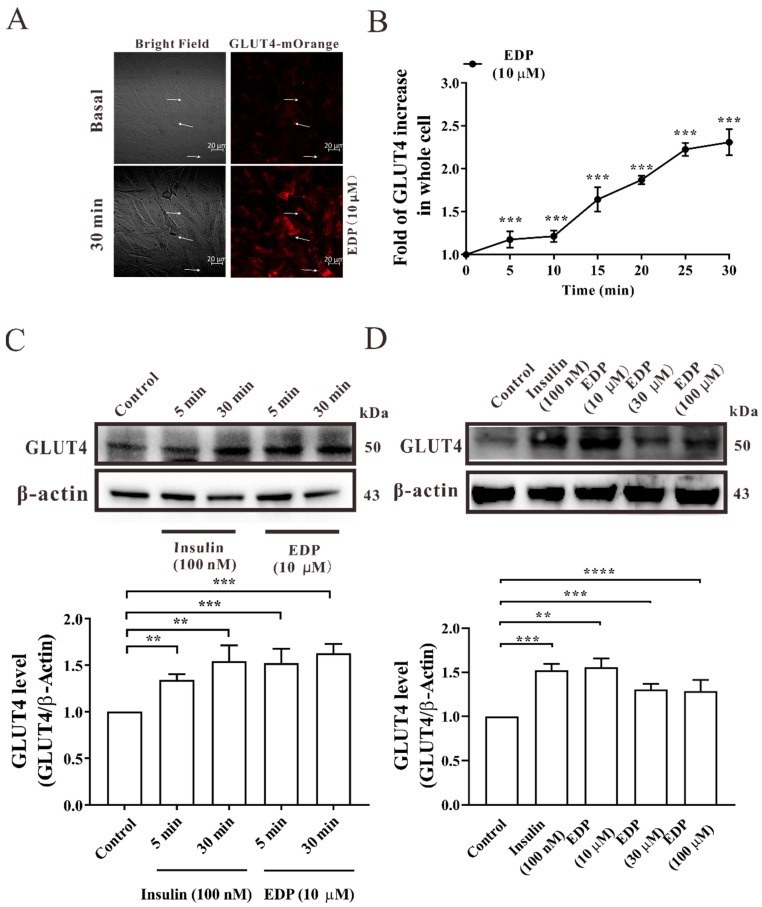
Estradiol Dipropionate (EDP) promotes GLUT4 expression. (**A**) Treatment with 10 μM EDP for 30 min significantly increases GLUT4 expression. Scale bar = 20 μm. (**B**) Images of EDP stimulating GLUT4 expression occurred in a time-dependent manner, n = 30 cells. (**C**) Effect of insulin and EDP treatment on GLUT4 protein expression in different time gradients. (**D**) The expression of GLUT4 induced by EDP at different concentration gradients was detected by Western blot. ** *p* < 0.01; *** *p* < 0.001; **** *p* < 0.0001.

**Figure 3 pharmaceuticals-16-00025-f003:**
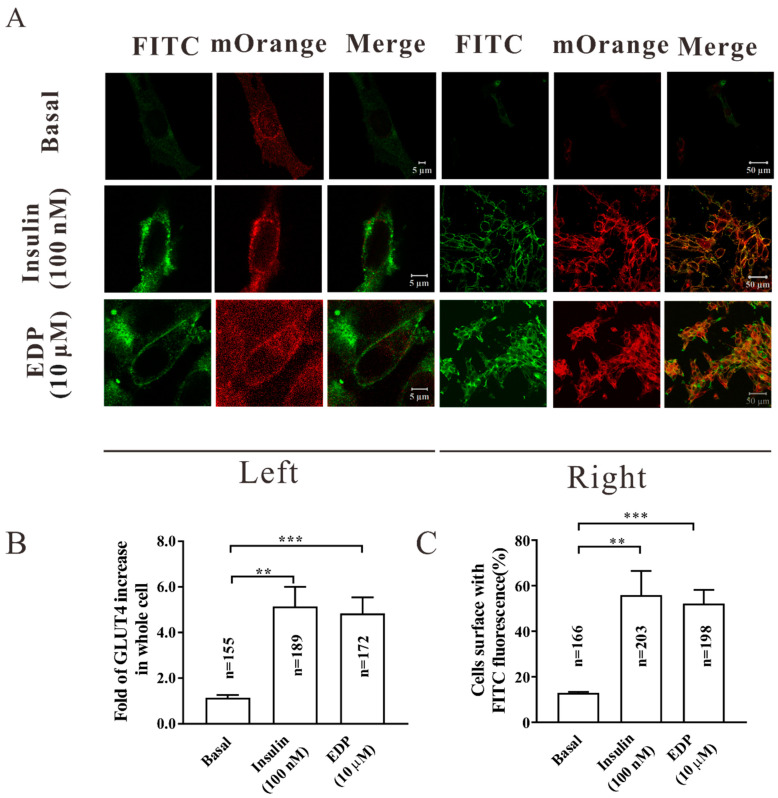
Estradiol Dipropionate (EDP) increased GLUT4 expression and fusion with the plasma membrane in L6 cells. (**A**) The amount of GLUT4-mOrange red fluorescence and FITC green fluorescence in L6 cells treated with 100 nM insulin or 10 μM EDP were detected by fluorescence assay within 30 min. On the left is a single cell. Scale bar = 5 μm. On the right is multicellular. Scale bar = 50 μm. (**B**) EDP with a concentration of 10 uM promoted the expression of GLUT4 in L6 cells within 30 min. (**C**) The fusion of GLUT4 with plasma membrane stimulated by EDP was quantified by FITC green fluorescent cells. The n value represents the number of GLUT4-mOrange cells in each group, data represent mean ± s.e.m. of values from three separate experiments with between 150 and 250 cells being examined in each experiment. ** *p* < 0.01; *** *p* < 0.001.

**Figure 4 pharmaceuticals-16-00025-f004:**
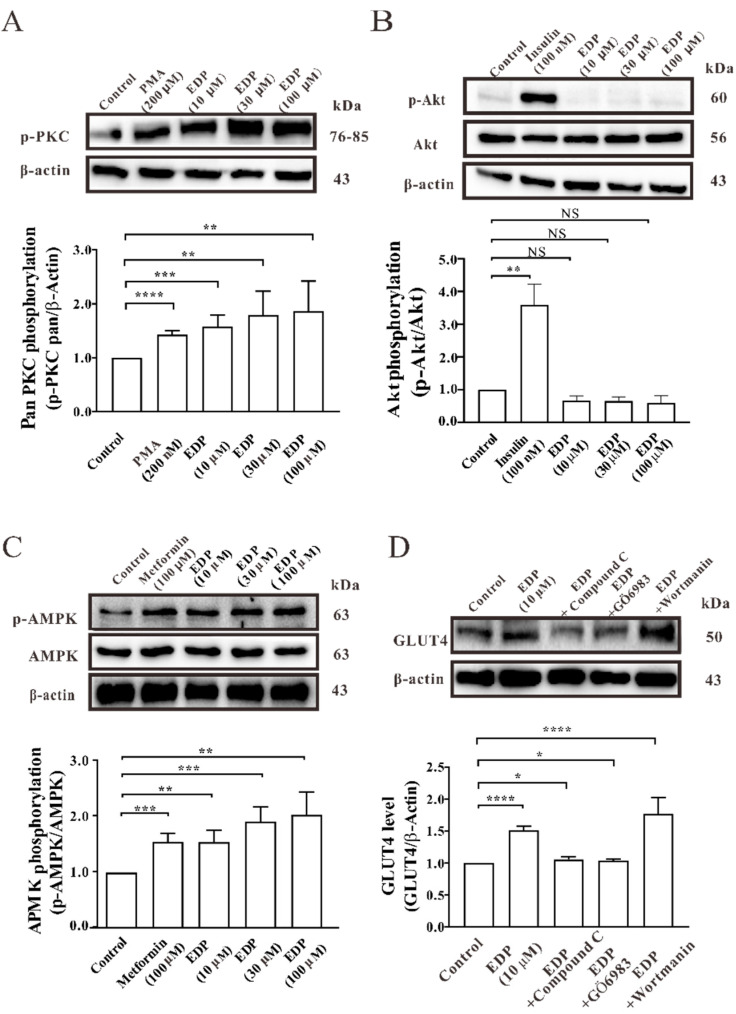
Estradiol Dipropionate (EDP) induces GLUT4 expression in L6 cells by promoting phosphorylation of the PKC and AMPK signaling pathways. (**A**) PMA and different concentrations of EDP significantly increased PKC phosphorylation levels after 30 min of stimulation. (**B**) The phosphorylation of Akt had no effect on L6 cells stimulated by EDP at different concentrations for 30 min. (**C**) The phosphorylation of AMPK increased significantly after 30 min of stimulation with 100 μM metformin and different concentrations of EDP. (**D**) The effect of 10 μM EDP-induced GLUT4 protein expression was blocked by two inhibitors, PKC inhibitor compound C and AMPK inhibitor Gö6983 for 30 min in L6 cells. NS *p* > 0.05; * *p* < 0.05; ** *p* < 0.01; *** *p* < 0.001; **** *p* < 0.0001.

**Figure 5 pharmaceuticals-16-00025-f005:**
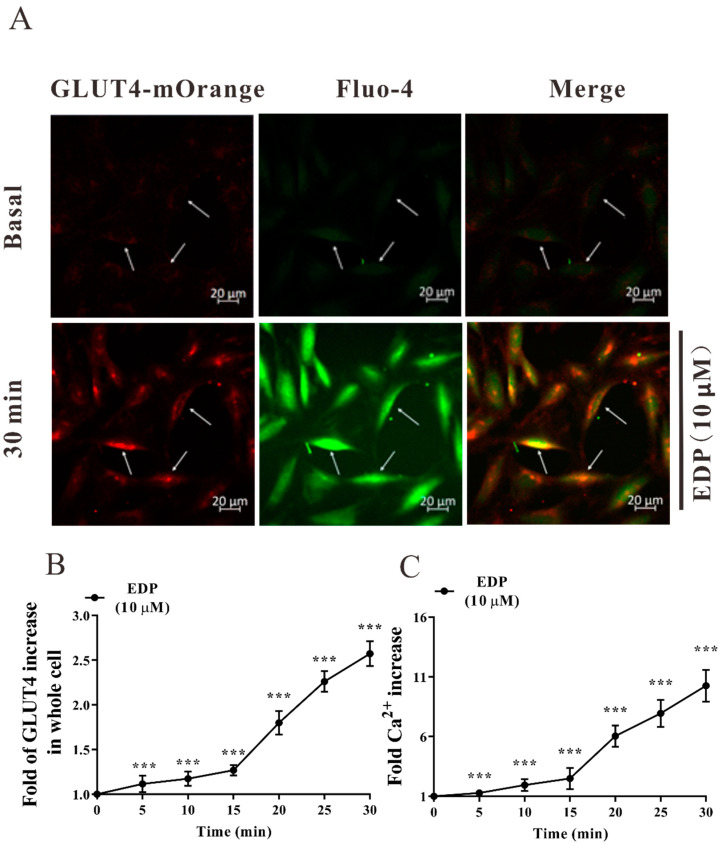
Estradiol Dipropionate (EDP) stimulated GLUT4 expression and intracellular Ca^2+^ level. (**A**) GLUT4, intracellular Ca^2+^ levels and overlays were specifically localized by GLUT4-mOrange, Flou4-AM indicator, and fluorescence Merge, respectively. Scale bar = 20 μm. (**B**) EDP-induced GLUT4 expression occurred in a time-dependent manner for 30 min, n = 30 cells. (**C**) The Ca^2+^ fluorescence intensity of L6 cells was calculated by ZEN 2010 software, n = 30 cells. *** *p* < 0.001.

**Figure 6 pharmaceuticals-16-00025-f006:**
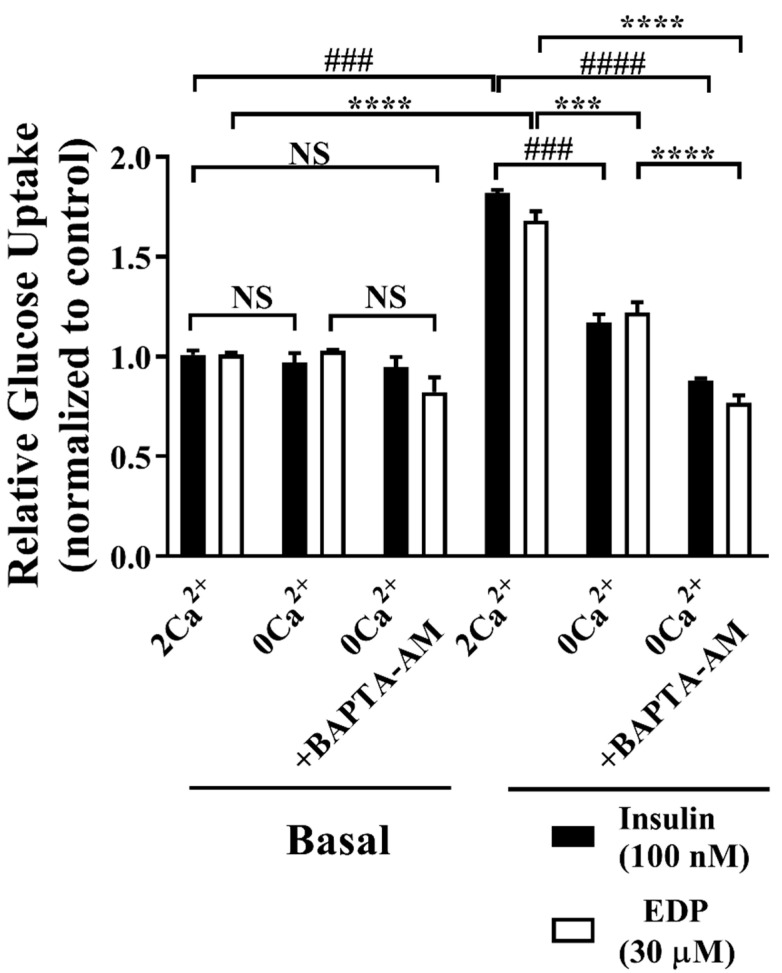
Intracellular Ca^2+^ promotes EDP-induced uptake of glucose. EDP-stimulated uptake of glucose in the presence of different concentrations of Ca^2+^ within 30 min. NS *p* > 0.05; ^###^
*p* < 0.001, ^####^
*p* < 0.0001 indicates comparison between blank and insulin group, and *** *p* < 0.001, **** *p* < 0.0001 indicates comparison between blank and EDP group.

**Figure 7 pharmaceuticals-16-00025-f007:**
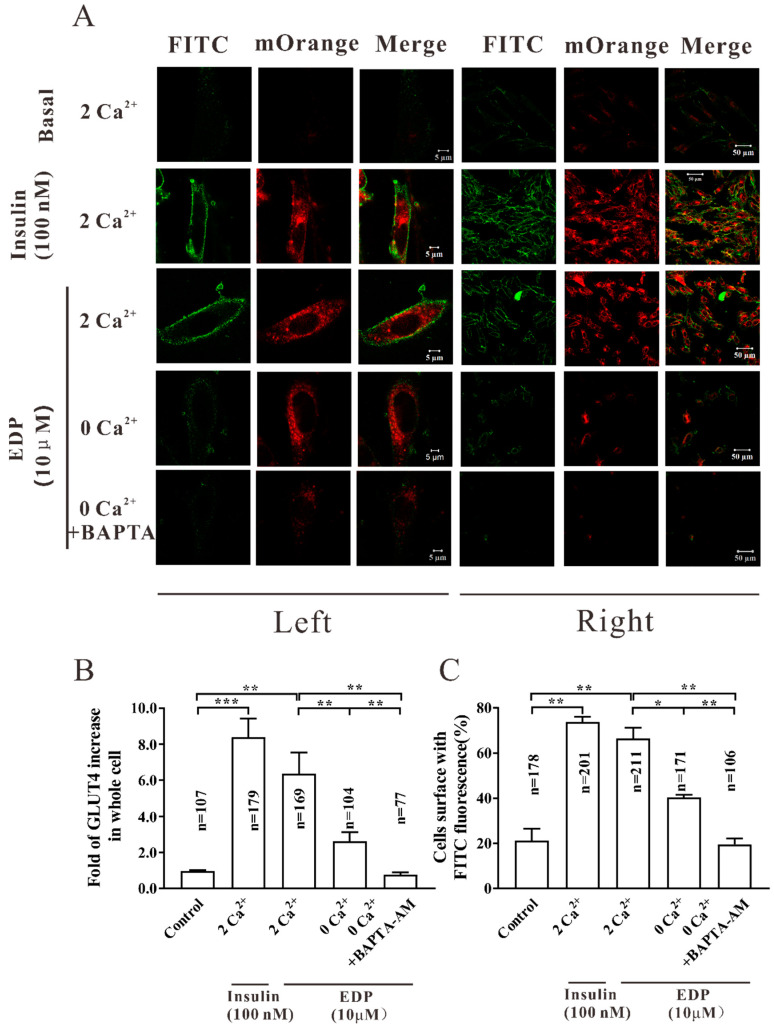
Ca^2+^ is involved in EDP-induced GLUT4 expression and plasma membrane fusion. (**A**) Images of 10 uM EDP induced-GLUT4 fusion into plasma membrane under different concentrations of intracellular Ca^2+^ level within 30 min. On the left is a single cell. Scale bar = 5 μm. On the right is multicellular. Scale bar = 50 μm. (**B**) Ca^2+^ mediated EDP to promote GLUT4 expression in L6 cells within 30 min. (**C**) Activated FITC fluorescence cell count and quantitation in L6 cells within 30 min. The n value represents the number of GLUT4-mOrange cells in each group, data represent mean ± s.e.m. of values from three separate experiments with between 50 and 250 cells being examined in each experiment. * *p* < 0.05; ** *p* < 0.01; *** *p* < 0.001.

**Figure 8 pharmaceuticals-16-00025-f008:**
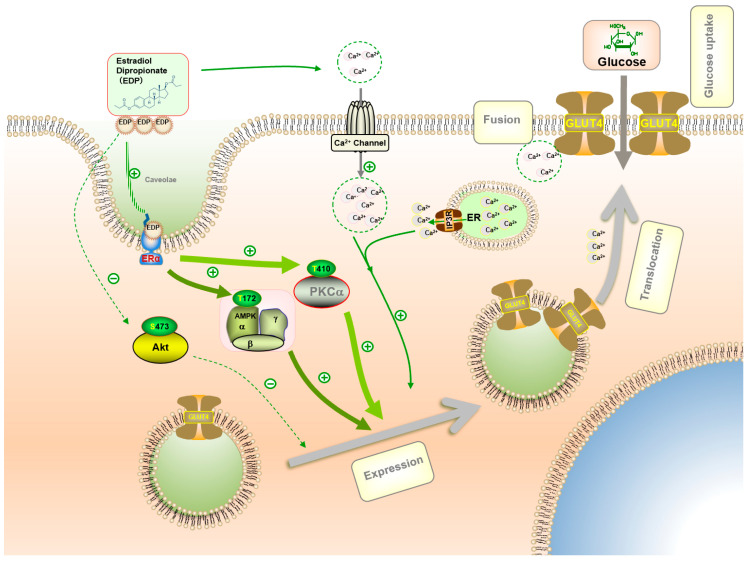
Estradiol Dipropionate (EDP) promotes the main pathways of uptake of glucose in L6 cells. EDP promotes GLUT4 expression through PKC and AMPK pathways, which in turn enhances GLUT4 fusion with the plasma membrane, while EDP also increases intracellular Ca^2+^ concentration in L6 cells, thereby performing uptake of glucose. Thus, both EDP-induced GLUT4 fusion into the plasma membrane and uptake of glucose were Ca^2+^ dependent.

## Data Availability

Data is contained within the article.

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
