# Peer review of "Glucose Uptake Is Increased by Estradiol Dipropionate in L6 Skeletal Muscle Cells"

_pharmaceuticals, 2022, doi:10.3390/ph16010025_

Round 1
Reviewer 1 Report
The current study investigated the mechanism of Estradiol Dipropionate (EPD) on the uptake of glucose in L6 skeletal muscle cells, and found that EDP promoted GLUT4 expression and transduction to accelerate glucose uptake. This study is interesting. However, there are some questions should to be clarify or solved.
1. Please point out the source of L6 skeletal muscle cells in the present study. Where did them obtained from?
2. Please clarify the Cat. Number or Lot. Number of all the reagents in the present study.
3. The accuricy of the glucose uptake assay is doubtful. Did the culturing medium contain phenol red? (since you use alpha-MEM from Gibco which might contain the substances). If it do contain phenol red, it might interfere with the absorbance value of red quinone mimide pigment produced by glucose oxidation.
4. Does the concentration of ions such as Ca2+ or pH of the culture medium solution interfere with the color reaction of the red quinone mimide pigment?
5. How did EDP stimulate the intracellular Ca2+ level (Figure 2A) ? Could it be associated with the AMPK or PKC signaling pathways? Please make an additional exploration to clarify the question.
6. Are there any EDP receptors on the surface of L6 skeletal muscle cells? Does the binding of EDP to its receptor activate GLUT4 expressioin continiously? How did EDP degrade to maintain a biological hemeostasis?
7. Figure 1B and 1C, Figure 6, "Glucose uptake" sholud be modified as "Relative glucose uptake (normalized to control)" for an accurate expression.
8. Introduction: "EDP is a hypoglycemic monomer drug previously screened in our lab from diabetes models", I do not see any data or reference cited to verify the statement.
9. Reference: I do not see the recent or newly progress in this field. The reference cited in the manuscript is relatively old.
Author Response
Response to Reviewer 1 Comments
- Please point out the source of L6 skeletal muscle cells in the present study. Where did them obtained from?
Response: The L6 cell line were purchased from Procell Life Science & Technology Company (Cat# CL-0136, Wuhan, China). The source of L6 rat myoblasts has been supplemented in line 312 of the manuscript.
- Please clarify the Cat. Number or Lot. Number of all the reagents in the present study.
Response: The Cat. Number or Lot. Number of all the reagents have been added in the chemicals and reagents section of the manuscript and in the protein western blotting section in this study.
- The accuricy of the glucose uptake assay is doubtful. Did the culturing medium contain phenol red? (Since you use alpha-MEM from Gibco which might contain the substances). If it do contain phenol red, it might interfere with the absorbance value of red quinone mimide pigment produced by glucose oxidation.
Response: Although phenol red does exist in alpha-MEM from Gibco, but quinone amide red color produced by glucose oxidase does not affect the absorbance value of EDP to cells. The first reason is that the color of phenol red has been excluded by the InTec.PRODUCTS.INC. when preparing the glucose oxidase kit. Only triglyceride, bilirubin, vitamin C and hemoglobin will affect the interference experiment; The most important thing is that when we set up the blank control, we processed two groups of blank controls. One was the control group that only added the culture medium without L6 cells, and the other was the negative control group that added DMSO (solvent) with L6 cells. In this way, when we processed the data, there would be no influence of the substances in the culture medium.
- Does the concentration of ions such as Ca2+ or pH of the culture medium solution interfere with the color reaction of the red quinone mimide pigment?
Response: As the third question says, during the production of the glucose oxidase kit manufacturer, the environmental factors such as Ca2+, PH in the medium were detected, but the reaction to the oxidase was not affected.
- How did EDP stimulate the intracellular Ca2+ level (Figure 2A)? Could it be associated with the AMPK or PKC signaling pathways? Please make an additional exploration to clarify the question.
Response: Under the condition of 2 mM Ca2+, EDP stimulated the intracellular calcium level (Figure 5 in the manuscript has been verified). It can be seen from Figure 4A and 4C that EDP promotes the phosphorylation level of PKC and AMPK. Under the conditions of 0 mM Ca2+ and 0 mM Ca2+ +BAPTA, the phosphorylation level of AMPK and PKC decreased significantly, speculating that calcium existed upstream of the pathway (Supplementary Figure 1). Due to the impact of the epidemic, only one experiment was performed in the supplementary Figure 1, which did not reach three replicates, so further confirmation could not be carried out.
Supplementary Figure 1 EDP-induced phosphorylation of AMPK and PKC is Ca2+-dependent in L6 cells.
- Are there any EDP receptors on the surface of L6 skeletal muscle cells? Does the binding of EDP to its receptor activate GLUT4 expressioin continiously? How did EDP degrade to maintain a biological hemeostasis?
Response: Estrogens are small lipophilic substances, which can directly penetrate into the nucleus or cytosol of target cells where they bind to estrogen receptors (ERα and ERβ) [1]. Estrogens exert their actions by binding to specific receptors, the estrogen receptors (ERs), which in turn activate transcriptional processes and/or signaling events that result in the control of gene expression (Supplementary Figure 2) [2]. EDP stimulates GLUT4 expression to be continuously expressed within 30 min, as shown in Figure 2 in the manuscript. There is no literature to explore relevant experiments, so we do not know how EDP degrades to maintain biological hemeostasis.
Supplementary Figure 2 Figure 7. Genomic and non-genomic estrogen signaling pathways. There are different estrogen-mediated signaling mechanisms. 1) Direct genomic signaling: estrogen binds to ERs. The complex dimerizes and translocate to the nucleus inducing transcriptional changes in estrogen-responsive genes with or without EREs. 2) Indirect genomic signaling: the membrane bound receptor induces cytoplasmic events such as modulation of membrane-based ion channels, second-messenger cascades and transcription factors. 3) ER-independent: estrogen exerts antioxidant effects in an ER-independent manner. 4) Estrogen independent: ligand-independent genomic events.
References
- Ikeda, K.; Horie-Inoue, K.; Inoue, S. Functions of estrogen and estrogen receptor signaling on skeletal muscle. J Steroid Biochem Mol Biol 2019, 191, 105375, doi:10.1016/j.jsbmb.2019.105375.
- Fuentes, N.; Silveyra, P. Estrogen receptor signaling mechanisms. Adv Protein Chem Struct Biol 2019, 116, 135-170, doi:10.1016/bs.apcsb.2019.01.001.
- Figure 1B and 1C, Figure 6, "Glucose uptake" sholud be modified as "Relative glucose uptake (normalized to control)" for an accurate expression.
Response: The relative glucose uptake (normalized to control) has been changed accordingly in Figures 1B and 1C, Figure 6 of the manuscript.
- Introduction: "EDP is a hypoglycemic monomer drug previously screened in our lab from diabetes models", I do not see any data or reference cited to verify the statement.
Response: This sentence was incorrectly described in the manuscript. In the previous work, we found a hypoglycemic monomer drug screened from the L6 cell diabetes model, which has been modified in the introduction of the manuscript in line 74. Data on glucose uptake and elevated GLUT4 expression in the L6 cell diabetes model manuscript, which is not yet available for publication.
- Reference: I do not see the recent or newly progress in this field. The reference cited in the manuscript is relatively old.
Response: Some of the references in the manuscript have been modified into articles of recent years.

Reviewer 2 Report
1. Authors should submit manuscripts in a suitable template for the journal. I would reconsider the manuscript afterwards.
2. Abbreviations in Figure Titles make it difficult and confusing. Authors should write the full term and abbreviate subsequently in each Figure title.
3. Abstract is very technical, and it must be simplified and should be written in full terms.
4. Figure 8 must be justified with the data from this manuscript. Otherwise, please remove it.
5. Accurate source of the cells should be clearly written, and how the nature of cells be characterised.
Author Response
Response to Reviewer 2 Comments
- Authors should submit manuscripts in a suitable template for the journal. I would reconsider the manuscript afterwards.
Response: The structure of the manuscript has been readjusted.
- Abbreviations in Figure Titles make it difficult and confusing. Authors should write the full term and abbreviate subsequently in each Figure title.
Response: The full terms and abbreviations in each Figure title have been supplemented in the manuscript.
- Abstract is very technical, and it must be simplified and should be written in full terms.
Response: The technical part of the abstract has been revised and simplified in the manuscript.
- Figure 8 must be justified with the data from this manuscript. Otherwise, please remove it.
Response: Figure 8 in the manuscript was re-improved, that is, the data in the manuscript was summarized.
- Accurate source of the cells should be clearly written, and how the nature of cells be characterised.
Response: Skeletal muscle myoblast cells of rat L6 were purchased from Procell Life Science & Technology Company (Cat# CL-0136, Wuhan, China). The accurate source of L6 cells has been supplemented in line 312 of the manuscript. The skeletal muscle is one of the largest organs in the human body and plays an important role in movement and systemic metabolic homeostasis. The formation of muscle fibers during myogenesis and muscle repair involves the activation of progenitor cells, which multiply in the form of mononuclear myoblasts and eventually fuse to form multinucleated myotubules, this complex arrangement limits the methods that can be used to study the processes within myoblasts in vivo. However, molecular exploration can be gained from in vitro models of skeletal muscle, such as L6 cells isolated from a newborn rat thigh muscle culture. The nature characteristics of L6 cells are supplemented in line 40 of the manuscript.
